# Fetal Tissue-Derived Mast Cells (MC) as Experimental Surrogate for In Vivo Connective Tissue MC

**DOI:** 10.3390/cells11060928

**Published:** 2022-03-08

**Authors:** Caterina Iuliano, Magdalena Absmaier-Kijak, Tobias Sinnberg, Nils Hoffard, Miriam Hils, Martin Köberle, Florian Wölbing, Ekaterina Shumilina, Nicole Heise, Birgit Fehrenbacher, Martin Schaller, Florian Lang, Susanne Kaesler, Tilo Biedermann

**Affiliations:** 1Department of Dermatology and Allergology, School of Medicine, Technical University Munich, Biedersteiner Str. 29, 80802 Munich, Germany; caterina.iuliano@tum.de (C.I.); magdalena.absmaier@tum.de (M.A.-K.); nils.hoffard@tum.de (N.H.); miriam.hils@tum.de (M.H.); martin.koeberle@tum.de (M.K.); florian.woelbing@tum.de (F.W.); 2Department of Dermatology, University of Tübingen, 72076 Tübingen, Germany; tobias.sinnberg@med.uni-tuebingen.de (T.S.); birgit.fehrenbacher@med.uni-tuebingen.de (B.F.); martin.schaller@med.uni-tuebingen.de (M.S.); 3Department of Physiology, University of Tübingen, 72076 Tübingen, Germany; ekaterina.shumilina@uni-tuebingen.de (E.S.); nicole.heise@freenet.de (N.H.); florian.lang@med.uni-tuebingen.de (F.L.)

**Keywords:** mast cell, in vitro model, transcriptome, RNA sequencing, proteases, IgE

## Abstract

Bone-marrow-derived mast cells are matured from bone marrow cells in medium containing 20% fetal calf serum (FCS), interleukin (IL)-3 and stem-cell factor (SCF) and are used as in vitro models to study mast cells (MC) and their role in health and disease. In vivo, however, BM-derived hematopoietic stem cells account for only a fraction of MC; the majority of MC in vivo are and remain tissue resident. In this study we established a side-by-side culture with BMMC, fetal skin MC (FSMC) or fetal liver MC (FLMC) for comparative studies to identify the best surrogates for mature connective tissue MC (CTMC). All three MC types showed comparable morphology by histology and MC phenotype by flow cytometry. Heterogeneity was detected in the transcriptome with the most differentially expressed genes in FSMC compared to BMMC being *Hdc* and *Tpsb2*. Expression of ST2 was highly expressed in BMMC and FSMC and reduced in FLMC, diminishing their secretion of type 2 cytokines. Higher granule content, stronger response to FcεRI activation and significantly higher release of histamine from FSMC compared to FLMC and BMMC indicated differences in MC development in vitro dependent on the tissue of origin. Thus, tissues of origin imprint MC precursor cells to acquire distinct phenotypes and signatures despite identical culture conditions. Fetal-derived MC resemble mature CTMC, with FSMC being the most developed.

## 1. Introduction

Mast cells (MC) are tissue-resident immune cells that play a crucial role in different types of immune responses [1]. MC arise from two distinct waves of migration, originating from both erythromyeloid progenitors from the extraembryonic yolk sac, as well as from hematopoietic stem cells [2]. Of note, postnatally in vivo, bone-marrow-derived hematopoietic stem cells account for only a fraction of MC in the body and mainly replenish the mucosal MC population [3]. Contrary to the classic tree-like model describing hematopoiesis, single-cell RNA sequencing studies suggest that hematopoietic stem cells and progenitor cells differentiate along trajectories [4]. These studies postulate that the MC developmental route is positioned adjacent to the erythrocyte and basophil trajectories and a common MC/basophil progenitor [5]. MC progenitors (MCP) migrate into target tissues during their maturation, i.e., the intestine, skin, or serosal cavity, by extravasating from the bloodstream under basal conditions dependent on trafficking molecules CXCR2 and ß7 integrin and chemoattractant gradients [6,7]. Their preferential localization in the host–environmental interface allows MC to act as sentinels for tissue damage and pathogen invasion [8]. The process of becoming mature MC following tissue infiltration as progenitor cells is controlled by the microenvironment of the tissue providing stimuli such as IL-3, that lead to maturation of MC [9].

Research on MC over decades has focused on the crucial role of MC in allergic diseases such as allergic asthma and delayed-type hypersensitivity [10,11]. More recently, it has become apparent that MC play a crucial role as effector cells in innate and other adaptive immune responses [12,13]). MC express several stimulatory and inhibitory surface receptors, allowing them to respond to external stimuli by finely tuned immune responses [14]. Among the activating receptors of MC are the high-affinity IgE receptor FcεRI, the Mas-related G protein-coupled receptor X2 (MRGPRX2), the alarmin IL-33 receptor ST2/IL33R and pattern recognition receptors (PRRs) sensing also danger signals, such as Toll-like receptors (TLRs). While mediators such as histamine and tryptase are stored preformed in granula and released upon degranulation, cytokines, chemokines, and growth factors are synthesized upon stimulation and released de novo [15].

In vivo models using constitutive or inducible MC-deficient mice have been very important and have helped gather compelling evidence for the relevance of MC in different physiological and pathological processes [12]. However, to study and fully understand MC specific functions on a cellular level, in vitro experiments are also necessary, creating a need for appropriate surrogates and models both of human and mouse origin. However, since MC mature within tissues under the direct influence of the local tissue microenvironment, it is crucial to define MC surrogates for in vitro studies more closely. Regarding mouse MC, BMMC are a helpful model for MC studies and frequently used, however, BMMC phenotypically and functionally differ from tissue-resident MC populations [16]. To be able to conduct in vitro studies, a method is needed that provides MC in sufficient quantity. Using tissue-resident MC from mature origins such as adult skin neither provides sufficient yield for in vitro analysis nor do mature MC proliferate in vitro, whereby MC from mature origin do not act as potential surrogates.

Over the past decades, various methods have been developed to promote the differentiation and maturation of MC from precursor cells and to effectively extract them from tissues [17]. However, transcriptional analysis of MC originating from different tissues has demonstrated considerably greater heterogeneity across tissues than previously appreciated [18]. Due to this increasing evidence of MC heterogeneity and tissue specificity, it is crucial to address this issue in experimental setups and to also choose the appropriate MC surrogate for in vitro studies. To this end, a better characterization and side-by-side comparison of MC surrogates could be a prerequisite for the most adequate selection. In the present study, we describe the generation of MC originating from bone-marrow (BMMC), fetal liver (FLMC), and fetal skin (FSMC) and compare these three MC types regarding phenotypic markers, proliferation, and mediator content as well as response to mitotic activation, IgE-crosslinking of the FcεRI receptor and innate stimulation. While histology and flow cytometric analyses revealed nearly identical results following standardized culture conditions, our data showed differences between the three MC types at the gene expression level and in regard to mediator release. Primarily due to high histamine content and strong reactivity towards type 2 immune signals, fetal-skin derived MC were identified as the preferred choice for analyses aiming to decipher the role of MC in allergic inflammation and atopic diseases, especially of the skin. 

## 2. Materials and Methods

### 2.1. Animals

C57BL/6N mice were purchased from Charles River (Sulzfeld, Germany). Animals were kept under specific pathogen-free conditions in accordance with the guidelines of the Federation of European Laboratory Science Association and the German Animal Welfare Act.

### 2.2. In Vitro Generation of Mast Cells

Generation of BMMC: 6- to 8-week-old C57BL6N mice were euthanized by isoflurane and cervical dislocation. Bone marrow cells were then prepared from femur and tibia and cultured in RPMI 1640 medium (Biochrom, Berlin, Germany) with 20% FCS (CH 30160.03 Hyclone Perbio), 1% X63Ag8-653mIL-3–conditioned medium corresponding to 10 ng IL-3/mL, 1% CHO–murine stem cell factor conditioned media corresponding to 50 ng mSCF/mL), 50 U/mL penicillin/streptomycin (Biochrom, Berlin, Germany) and 20 µM β-mercaptoethanol (Merck, Darmstadt, Germany) (henceforth referred to as MC-medium), for 21 days at 37 °C and 5% CO_2_. Conditioned media were obtained from cell culture supernatants of the above-mentioned cell lines. Concentration estimation of IL-3 and SCF was performed using proliferation assays with FDCP-1 cells (IL-3 dependent) and TF-1 (SCF-dependent).

Generation of FSMC and FLMC: Fetuses were retrieved from C57Bl/6N mice 16 days post gestation period and decapitated. Back skin was prepared from the fetuses, digested in 0.25% Trypsin-EDTA solution (Gibco Thermo Fisher Scientific, Munich, Germany) and passed through cell strainers (Cambrex, NJ, USA). The livers were removed from fetuses and single cell suspensions were prepared from them. Erythrocytes were lysed with ACK lysis buffer (BioWhittaker, Lonza, Swiss) and cell suspension was cultured for 21 days in complete mast-cell medium as described before. All cultured media were tested regularly for mycoplasma and no contamination was detected.

### 2.3. Flow Cytometry for Mast Cell Surface Markers

The surface expressions of FcεRI, CD117 (c-kit), ST2, β-Integrin and CD49b were analyzed by flow cytometry. Briefly, 1 × 10^6^ mast cells were washed with PBS and incubated with an anti-CD16/CD32 mAb (clone 32, Biolegend, London, UK) for Fc receptor blocking. The following primary, fluorochrome-coupled antibodies were used for analysis: anti-CD45.2 (clone 104, Biolegend), anti-FcεRI (clone MAR-1, Biolegend), anti-CD117 (clone 2B8, Biolegend), anti-ST2 (clone DIH4, Biolegend), anti-integrin-ß7 (clone FIB27, Biolegend) and anti-CD49b (clone DX5, Biolegend) and Live/Dead fixable aqua dead cell stain (Invitrogen Thermo Fischer, Munich, Germany). Cells were determined fully matured if >90% were FcεRI^+^CD117^+^ and only then further studies were conducted.

### 2.4. Histology

Histological staining was performed after cells were determined matured as determined by flow cytometry. A total of 1 × 10^4^ cells were cytocentrifuged onto a glass slide and air dried. For Alcian Blue staining, slides were stained with 1.0% alcian blue (Sigma-Aldrich/Merck, Darmstadt, Germany) at pH 2.5 in 1% acetic acid, followed by staining for 15 min with 0.1% safranin O at 22 °C (Sigma-Aldrich/Merck, Darmstadt, Germany). For Giemsa staining, slides were stained in 5% Giemsa (Sigma-Aldrich/Merck) for 20 min at 22 °C. Slides were examined under a light microscope (DM3000, Leica Biosystems; Nußloch, Germany).

### 2.5. Transmission Electron Microscropy

Cells were fixed with Karnovsky’s fixative for 24 h at 4 °C. Post-fixation was based on 1.0% osmium tetroxide containing 1.5% K-ferrocyanide in 0.1 M cacodylate buffer for 2 h. After following standard methods, blocks were embedded in glycide ether and cut using an ultramicrotome (Ultracut, Reichert, Vienna, Austria). Semithin sections were stained with Toluidinblue (1% Morphisto, Frankfurt am Main, Germany) and examined under a Nikon eclipse 80i microscope (Nikon Europe BV, Amsterdam, The Netherlands). Ultra-thin sections (30 nm) were mounted on copper grids and analyzed using a Zeiss LIBRA 120 transmission electron microscope (Carl Zeiss, Oberkochen, Germany) operating at 120 kV.

### 2.6. Proliferation, Histamine Release and Hexosaminidase Content

For proliferation experiments, MC were cultured with 0.25 µCi of [^3^H] Thymidine (Amersham, GE Healthcare, Freiburg, Germany) for 14 h. The cells were harvested using a FilterMate Harvester (PerkinElmer LAS, Rodgau, Germany) and incorporation analyzed in a MicrobetaTriLux Luminescence Counter (PerkinElmer LAS, Rodgau, Germany). Incorporated radioactivity is expressed as counts per minute.

For IgE sensitization, 1 × 10^5^ cells were pre-incubated with 1µg/mL anti-DNP IgE (Sigma-Aldrich/Merck) for 24 h at 37 °C. For cross-binding and activation of the receptor, the cells were then stimulated with 40 ng/mL DNP-HSA (Sigma-Aldrich/Merck) for 30 min. Control stimulation was performed with phorbol-12-Myristat-13-Acetate (PMA) (Sigma, Taufkirchen, Germany) and ionomycin (Sigma-Aldrich/Merck) or 0.1% TritoxX100 (Sigma-Aldrich/Merck) for complete lysis at 37 °C. For histamine ELISA, cells were cultured in MC-medium, while for ß-hexosaminidase assay, cells were transferred in Tyrodes Buffer (Sigma). Histamine release was analyzed using Histamine ELISA (LSBio, Seattle, WA, USA). For ß-hexosaminidase release, 50 µL of supernatant was incubated with 50 µL of p-nitrophenyl-N-acetyl-β-d-glucosaminide (Sigma-Aldrich/Merck) for 2 h at 37 °C. Reaction was stopped with Tris-HCl buffer (Sigma-Aldrich/Merck) and colorimetric measured at 405 nm using an ELISA Reader Multiskan EX (Thermo Fisher Scientific, Munich, Germany).

### 2.7. Cytokine Measurements

A total of 1 × 10^6^ MC were cultured in 24-well plates and stimulated with either 100 ng/mL Pam2Cys (EMC microcollections, Tübingen, Germany), 100 ng/mL Pam3Cys (EMC microcollections, Tübingen, Germany), 10 ng/mL LPS from *s.minnesota* (Alexis, Lausen, Swiss), 0.5 µM CpG OD 1668 (Eurofins MWC Operon, Ebersberg, Germany), 10 ng/mL PolyIC (Invivogen, Toulouse, France), 10 ng/mL imiquimod (Invivogen), 10 ng/mL IL-33 (PeproTech, Hamburg, Germany) ionomycin (Sigma-Aldrich/Merck) or phorbol-12-myristat-13-acetate (PMA) (Sigma-Aldrich/Merck) for 24 h at 37 °C in MC-medium. Supernatants were quantified by enzyme-linked immunosorbent assay for IL-10 (R&D Systems, Minneapolis, MN, USA, detection limit 15.6–1000 pg/mL), IL-6 (R&D Systems, Minneapolis, MN, USA, detection limit 15.6–1000 pg/mL), TNF-a (R&D Systems, detection limit 10.9–700 pg/mL), LegendPlex Mouse Inflammation panel (BioLegend, San Diego, CA, USA) and LegendPlex Mouse Th Cytokine panel (Biolegend).

### 2.8. Patch-Clamp Analysis

Patch-clamp experiments were performed at room temperature in voltage-clamp, fast whole-cell mode. MC were continuously superfused by a flow system inserted into the dish. The bath was grounded via a bridge filled with NaCl Ringer solution. Borosilicate glass pipettes (2- to 4-megaohm (Mohm) tip resistance; GC 150 TF-10; Clark Medical Instruments) manufactured by a microprocessor-driven DMZ puller (Zeitz) were used in combination with a MS314 electrical micromanipulator (MW; Märzhäuser). The currents were recorded by an EPC-9 amplifier (HEKA) using Pulse software (HEKA) and an ITC-16 Interface (Instrutech). Whole-cell currents were determined as 10 successive 200-ms square pulses from a −35 mV holding potential to potentials between −115 mV and +65 mV. The currents were recorded with an acquisition frequency of 10 and 3 kHz low-pass filtered. Where indicated, the Ag DNP-HSA (50 ng/mL), endothelin-1 (100 nM), the channel blocker TRAM-34 (300 nM; Sigma, Taufkirchen, Germany), and/or the Ca^2+^ ionophore ionomycin (1 μM; Sigma) were added to the bath solution.

The offset potentials between both electrodes were zeroed before sealing. The original whole-cell current traces are depicted without further filtering and currents of the individual voltage square pulses are superimposed. The applied voltages refer to the cytoplasmic face of the membrane with respect to the extracellular space. The inward currents, defined as flow of positive charge from the extracellular to the cytoplasmic membrane face, are negative currents and are depicted as downward deflections of the original current traces.

### 2.9. Intracellular Calcium Measurement

Intracellular Ca^2+^ measurements were performed using fura-2AM. Briefly, MC were loaded with fura-2AM (2 μM; Molecular Probes) for 20 min at 37 °C. The cells were continuously superfused to remove leaked-out extracellular fura-2. Fluorescence measurements were conducted with an inverted phase-contrast microscope (Axiovert 100; Zeiss, Oberkochen, Germany). Cells were excited alternatively at 340 and 380 nm and the light was deflected by a dichroic mirror into either the objective (Fluar ×40/1.30 oil; Zeiss) or a camera (Proxitronic) mounted on the microscope. Emitted fluorescence intensity was recorded at 510 nm and data acquisition was performed by using Metafluor computer software (Universal Imaging). To determine the fluorescence intensity in single cells, the regions of interest were determined as circuits closely surrounding individual cells. After a control period to wash the cells and remove any non-gathered dye, intracellular Ca^2+^ was measured before and following addition of DNP-HSA (50 ng/mL) to IgE-sensitized MC, in the absence or presence of extracellular Ca^2+^. As a measure of the increase of cytosolic Ca^2+^, the slope and peak of the changes in the 340/380 nm ratio were calculated for each experiment. To reach nominally Ca^2+^-free conditions, experiments were performed using Ca^2+^-free Ringer solution containing (in mmol/L): 125 NaCl, 5 KCl, 1.2 MgSO_4_, 2 Na_2_HPO_4_, 32.2 Hepes, 0.5 EGTA, 5 glucose (pH 7.4). For intracellular calibration purposes, ionomycin (10 μM) was applied at the end of each experiment.

### 2.10. Analysis of Gene Expression

Total RNA was extracted from MC using the PeqGold Total RNA Kit (PeqLab, VWR, Darmstadt, Germany). A total of 1 µg of RNA was digested with the DNase I Kit according to the manufactures protocol (DNase I AMP Grade, Thermo Fischer) and cDNA synthesis performed using the iScript cDNA Synthesis Kit (Bio-Rad Laboratories, Munich, Germany) with 10 mM dNTPs, oligo(dt)-18 primer and reverse transcriptase according to the manufacturer’s instructions. Quantitative real-time PCR was carried out on a StepOne^TM^ Real-Time PCR System (LifeTechnologies GmbH, Darmstadt, Germany). Values were normalized to the housekeeping gene β-actin. A list of used primers is provided in the Appendix A.

### 2.11. RNA Sequencing

Total RNA was isolated from MC after four weeks of culture with the PeqGold Total RNA Isolation Kit (VWR, Radnor, PA, USA). Quality and integrity of total RNA was controlled using NanoDrop 2000 spectrophotometer (Thermo Fisher Scientific) and 2100 Bioanalyzer (Agilent Technologies). To prepare a library of barcoded cDNA for bulk 30-sequencing, poly(A)-mRNA was reverse transcribed with Maxima RT polymerase (Thermo Fisher Scientific, Munich, Germany) using oligo-dT primer containing barcodes, unique molecular identifiers (UMIs) and an adaptor. The 5′ ends of the cDNAs were extended by a template switch oligo (TSO) and full-length cDNA was amplified with primers binding to the TSO-site and the adaptor. NEB UltraII FS kit was used to fragment cDNA. After end repair and A-tailing, a TruSeq adapter was ligated and 3′-endfragments were finally amplified using primers with Illumina P5 and P7 overhangs. The library was sequenced on a NextSeq 500 (Illumina).

High-throughput gene expression data analysis was carried out using the R environment for statistical computing. Briefly, genes were normalized using DeSeq2 and a principal component analysis was performed. Next, pairwise comparisons were performed using edgeR and gene set enrichment analysis (GSEA). Additional gene set analysis was performed using the Degust webtool.

### 2.12. Statistics

If not otherwise stated, data are presented as the mean ± SEM. Statistical analysis of data was performed with unpaired Student’s *t* test (2 tailed) or with 2-way repeated-measures ANOVA and Tukey’s multiple-comparisons test. Analysis was performed with GraphPad Prism software (GraphPad, San Diego, CA, USA). *p* values less than 0.05 were considered statistically significant and are marked as follows: * *p* < 0.05; ** *p* < 0.001; *** *p* < 0.005.

## 3. Results

### 3.1. MC Phenotypes following Cultures of Bone Marrow Cells and Fetal Tissues

To generate MC in vitro, we isolated cells from bone marrow of adult C57BL/6N mice or fetal liver and fetal skin from C57BL/6N d16 embryos. We cultivated them in medium containing murine interleukin (mIL-)-3 and murine stem cell factor (mSCF) (Appendix A). The SCF receptor, c-kit (CD117), is one of the most critical receptors on mature MC, as a reduction in c-kit signaling leads to MC deficiency [6]. In line with this, our experiments demonstrated a downregulation of c-kit as early as 24 h after deprivation of SCF from culture medium (Appendix A). For comparability of the following analyses, culture conditions were kept constant for MC cultures from the three tissues investigated, i.e., bone marrow, fetal liver, and fetal skin, using medium supplemented with 20% FCS, 10 ng/mL IL-3 and 50 ng/mL SCF (Appendix A).

We first determined whether this in vitro culture was capable of generating MC from the different tissues. Flow cytometry analysis showed a difference in granularity, with FSMC showing the highest granularity (Appendix A). We classified MC from our cultures as fully developed MC by flow cytometry when >90% of viable cells were CD45.2^+^CD117^+^FcεRI^+^ (Figure 1A). Surface expression of integrin β7 on fetal MC precursors (MCP) is essential for the migration to the developing organ [19]. Committed MCP lose β7 integrin upon becoming MC [6]. Consistent with MC, the homing receptor β7 integrin declined in the cells from all three tissue types from week three of culture onwards, and was fully downregulated by week 5 (Appendix A). Fetal skin-derived MC were previously characterized as CD45^+^CD49b^+^CD117^+^FcεRI^+^, thereby showing a strong similarity to mature connective tissue (CT) MC [20,21]. Indeed, we found CD49b to be only expressed on fetal tissue-derived MC, suggesting that fetal-derived MC closely resemble mature cutaneous MC (Appendix A) [20,21]. Both BMMC and FSMC were fully developed after four weeks of culture, while FLMC required a longer time (see Table 1).

MC are mononuclear cells identified by metachromatic staining of their secretory granules. A well-established method for detecting secretory granules is toluidine blue staining, which stains MC in dark violet compared to the orthochromatic staining of other cell types [22]. Toluidine blue binds heparin, which is more abundant in mature connective tissue MC and correlates with the degree of differentiation towards mature MC [23]. The dark violet staining identified the generated cells of all three cultures as MC with more granules visible in FSMC compared to BMMC (Figure 1B, upper panel). This is confirmed by Giemsa staining, another method to identify MC by staining the cytoplasm of MC with dark blue color, while the granules are red-stained [24]. Giemsa staining of the MC confirmed the higher granularity of FSMC compared to BMMC (Figure 1B, middle panel). The combined staining of MC with Alcian blue and Safranin-O allows simultaneous differentiation of mucosal-like MC (MMC) and connective-tissue-like MC (CTMC) [25]. FSMC granules showed a more intense red coloration than BMMC, indicating a higher heparin content (Figure 1B, lower panel). When examined by electron microscopy, more numerous granules were seen in FSMC with the granules being larger and more homogeneous (representative pictures in Figure 1C). A cross section of the cell was analyzed and the number of granules per section quantified, showing that FSMC have the highest granule density (Figure 1D). In accordance with the lower yield, the lowest proliferation rate was found for FSMC (Figure 1E), indicating that MC maturity inversely correlated with proliferative capacity, as has also been shown for other cell types such as beta-cells [26] or cardiomyocytes [27]. Thus, all three tissues, bone marrow, fetal liver, and fetal skin, give rise to MC as seen in our side-by-side cultures, but histological staining and morphological properties indicate that FSMC are closest to tissue-derived mature MC whereas BMMC represent MC that are least developed/matured.

### 3.2. ST2 Expression on the Different MC and Response to IL-33 Stimulation

MC, together with ILC2 and Tregs, are the only cell types that constitutively express ST2, the receptor for IL-33, at high levels, while all other cell types that respond to extracellular IL33 are either ST2 negative at steady-state and only induce ST2 expression upon activation [28]. Therefore, as another marker for MC, we measured the surface expression of ST2 on BMMC, FLMC and FSMC. All three MC types express ST2 on the cell surface following our MC culture, however, FLMC showed a significantly reduced MFI for ST2 compared to BMMC, suggesting reduced density of ST2 on the surface of FLMC (Figure 2A,B). IL-33, an alarmin involved in type 2 immune responses, causes the activation and proliferation of MC through interaction with ST2 [29]. Thus, MC were stimulated with 10 ng/mL IL-33 for 1 h, cell pellets were harvested for RNA isolation and for further RT-PCR analysis. Upon stimulation with IL-33, BMMC and FSMC showed a comparably high increase in fold change for *Il13,* a MC-derived type 2 immune cytokine, while FLMC showed a significantly lower expression (Figure 2C). This result confirmed others e.g., by Leyva-Castillo et al., who showed IL-33-dependent upregulation of *Il13* expression in skin MC [30]. To test the functional relevance of the ST2 expression further, MC were stimulated with 10 ng/mL IL-33 for 24 h and type 2 specific cytokines were measured in the supernatant. Correlating surface expression of ST2, both BMMC and FSMC show a strong response to ST2-dependent activation by secreting high concentrations of type 2 immune cytokines such as IL-4, IL-9, IL-13 but also of IL-17, amongst others. In FLMC, which showed the lowest surface expression of ST2, only IL-10 and IL-9 were detectable in the supernatant upon IL-33 stimulation (Figure 2D).

### 3.3. Transcriptional Profiling of MC Derived from Cultures from Different Tissues

Differences in phenotypes, constitutive mediator production and granule contents should be associated with differentially regulated gene transcription. Thus, in the next step, we performed transcriptional profiling of the three MC types generated from the different tissues in vitro. For this purpose, RNA was isolated from 3 × 10^7^ cells from the cultured BMMC, FSMC, and FLMC, all of which were >90% CD117^+^FcεRI^+^ based on flow cytometry. Following standard procedure, RNA sequencing was performed and transcriptome profiling analyzed using the R-tool DESeq2 and the webtool Degust (www.degust.erc.monash.edu/degust/compare; accessed 12 December 2021). Principal component analysis revealed that the three MC populations cluster separately, demonstrating that differences in phenotype, morphology and behavior as detected in our previous experiments were indeed linked to alterations in gene transcription. Interestingly, MC derived from fetal skin (FSMC), represented in orange, showed the highest degree of variance compared to the other MC types (Figure 3A). Since the classic and most used method of MC generation in vitro is from bone marrow, we next normalized the expression levels of transcribed genes from FLMC and FSMC against those from BMMC.

MC of both the connective tissue (CTMC) and the mucosal type (MMC) store large numbers of cytoplasmic granules that are rapidly exocytosed following activation and which contain, among others, proteases, primarily serine proteases. CTMC and MMC differ in the expression of proteases [31]. Differential mRNA expression between the three MC types showed no significant difference in expression of CTMC-associated signature genes such as the serine protease *Mcpt5* (*Cma1*) and carboxypeptidase (*CPA*)3 in fetal-derived MC compared to BMMC, indicating that all three MC types can be classified as CTMC based on protease expression. This is further supported by RT-PCR data showing both Mcpt1 and Mcpt2, two typical mucosal type MC genes, not detectable in any of the MC types (Figure 3B). Analysis of the fetal MC types showed an increased fold change of *Camp* expression, which is associated with calcium-binding and IP3 signaling, in FLMC compared to BMMC (Figure 3C). Of note, fetal skin MC showed higher expression levels of mas-related G protein-coupled receptor (*Mrgpr*)*b2* and *Mrgprb1* (Figure 3D). Figure 3E depicts a heatmap of the top 20 differentially expressed genes within BMMC, FLMC and FSMC. The highest divergence in gene expression between BMMC and the fetal-derived MC show *Tpsb2*, the gene encoding for beta-II tryptase (Mcpt6) and *Hdc*, which catalyzes the biosynthesis of histamine from histidine. Both genes encode proteins characteristic for MC [32]. Both *S100a8* and *S100a9*, which are calcium-binding proteins that regulate inflammatory processes and immune responses towards TLR4 signaling, showed lower expression in FSMC.

Dywer et al. analyzed the heterogeneity of MC on a transcriptional level and revealed three distinct connective tissue MC subsets with varying capacity for in situ proliferation in the absence of tissue inflammation. Based on these data, a MC-specific gene signature was extrapolated [18]. Analyzing the gene expression of the Dwyer MC-signature in our three MC types showed a differential gene expression between the fetal-derived MC and BMMC (Figure 3F).

### 3.4. Consequences of IgE and Antigen Mediated Crosslinking of FcεRI in the MC Types Derived from Different Tissues

One important pathway to activate MC is crosslinking the FcεRI by IgE-antigen complexes, which leads to degranulation of MC [33]. We first looked at transcriptional differences in FcεRI signaling in the MC from the three tissue types. For this we performed RNA sequencing from unstimulated MC and analyzed differential gene expression of genes involved in FcεRI signaling, based on the pre-defined KEGG gene list (04664). As outlined above, we again normalized the expression levels of the genes of FLMC and FSMC against those of BMMC. Data are presented as a heatmap showing that, especially for genes related to the PI3K/Akt and MAPK signaling pathway, a log fold change of +3 in FSMC compared to BMMC was detected (Figure 4A). The transcriptional difference in genes involved in PI3K signaling in the BMMC, FLMC and FSMC, even in the absence of specific stimulation, supported our hypothesis that FSMC more closely resemble fully developed MC, as elevated PI3K activity drives MC maturation, while reduced PI3K activity modulates the maturation of in vitro MC towards a more myeloid-like cell type [34].

IgE-dependent activation of MC via FcεRI crosslinking activates the downstream phospholipase C-γ1cascade. The activation of protein kinase C releases calcium (Ca^2+^) from intracellular stores, followed by an influx of Ca^2+^ from the extracellular space through calcium channels. Ca^2+^ influx is indispensable for MC activation and degranulation and critically depends on the membrane potential [35,36]. This downstream activation of the IP3 cascade is essential for granulogenesis and the release of preformed mediators [37].

MC were loaded with 2 µM fura-2AM for 20 min at 37 °C and intracellular Ca^2+^ was measured before and following the addition of DNP-HSA (50 ng/mL) to IgE-sensitized MC in the absence or presence of extracellular Ca^2^. All three MC types showed unequivocal FcεRI-mediated Ca^2+^ influx (Figure 4B) and comparably sustained activation of the potassium channel SK4 essential for IgE-dependent MC degranulation [38] as demonstrated by patch-clamp analysis (Figure 4C). Consistent with our phenotype analyses and culture selection based on MC FcεRI expression, these data show that IgE-antigen complexes can activate all three MC populations.

Therefore, in the next step, we investigated mediators released in response to activation. Degranulation of MC was induced either by FcεRI crosslinking or by PMA/ionomycin of 1 × 10^6^ cells from each MC type in 1 mL volume. After 30 min, we collected the supernatants and measured histamine concentration by ELISA. As shown in Figure 4D, histamine release of FSMC was significantly higher compared to that of FLMC and BMMC. Interestingly, measuring histamine in the supernatant following stimulation with PMA/ionomycin, bypassing IgE-antigen-mediated FcεRI crosslinking, showed a significantly higher release of histamine from FSMC compared to BMMC, but also FLMC released large amounts (Figure 4D). The significant difference in histamine release is in line with the higher expression of *Hdc* and *Tpsb2* in FSMC and it corresponds to the higher granule content shown by electron microscopy for FSMC compared to BMMC and FLMC. The determination of β–hexosaminidase, another enzyme stored in MC granules and released upon degranulation, further confirmed our findings with fetal tissue-derived MC releasing higher concentrations of β–hexosaminidase (Figure 4E). Taken together, the higher granule content, the stronger response upon FcεRI signaling and the significantly higher release of histamine and ß-hexosaminidase from FSMC indicate a difference in MC development dependent on the tissue of origin.

### 3.5. Toll-like Receptor Expression and Activation of In Vitro Generated MC of Different Tissue Origin

In accordance with their role as innate sentinels, MC can also be activated through pathogen recognition receptors (PRR), including Toll like receptors (TLR) [39,40,41]. Thus, we first analyzed TLR expression by performing RT-PCR of unstimulated BMMC, FLMC and FSMC. Expression levels of the different TLRs relative to housekeeping gene β-actin were similar between the MC types for TLR3, TLR6, TLR7 and TLR9. Consistent with previous publications [42], no TLR5 expression could be detected (data not shown). Expression of TLR1, TLR2, and TLR4 was seen in all three MC types, but it was lower in FLMC (Figure 5A).

Next, 1 × 10^6^ cells of the three MC types were stimulated with Pam2Cys (TLR2/6), Pam3Cys (TLR1/2), poly:IC (TLR3), LPS (TLR4), Imiquimod (TLR7) or CpG OD 1668 (TLR9) for 24 h and concentrations of IL-6, known to be produced by MC upon activation, were measured in the supernatants. Interestingly and in contrast to the histamine release upon antigen-IgE activation, FSMC secreted the lowest amounts of IL-6 into the supernatant, while BMMC and FLMC showed higher levels of IL-6 production also compared to positive control stimulation by PMA/ionomycin. IL-6 upon TLR7 and TLR9 activation could only be measured in BMMC (Figure 5B). Regarding TNF-α production, TLR activation resulted in lower production in FSMC compared to BMMC and FLMC, especially in response to stimulation with Pam3Cys, LPS, and CpG (Figure 5C). Signaling of TLR4, the receptor for lipopolysaccharides (LPS) of Gram-negative bacteria, has been thoroughly investigated in MC and leads to NF-kB-dependent production of pro-inflammatory cytokines such as TNF-α and IL-6 [12,43]. The heatmap concentrations of inflammatory cytokines in the supernatant of BMMC, FLMC and FSMC upon stimulation with LPS for 24 h revealed a broad secretion of inflammatory cytokines from LPS-stimulated BMMC, with fewer cytokines being secreted by FLMC and FSMC, the latter only secreting TNF-α, MCP-1, and IL-6 (Figure 5D). FSMC are less responsive to TLR activation regarding cytokine production compared to FLMC and BMMC, the latter showing highest levels of TNF-α production upon LPS and CpG activation but again lowest release of histamine upon activation via FcεRI.

## 4. Discussion

MC biology and function have been fascinating many researchers over the years due to their multifaceted role in health and disease. Due to their plasticity, MC can act as direct effector cells or recruit innate and adaptive immune cells to modulate the immune response. Research focusing on MC has shown their essential role in allergy, autoimmune inflammation, and cancer [12,44,45]. In a recent paper, we were able to show an anti-tumorigenic effect of MC in the context of melanoma. Through activation of MC via TLR4, MC secrete CXCl10, which recruits tumor-specific T cells for effective antitumor responses [43]. Consequently, investigations on the role of MC steadily increase, involving both in vitro and in vivo analyses. As it is difficult to obtain primary MC in large numbers, it is well accepted to work with MC generated in vitro. In murine studies, classically bone marrow cells are used as input cells for these cultures, giving rise to large numbers of MC displaying many features of bona fide MC. However, these MC can only serve as models for mature tissue-type MC based on some shortcomings [46]. As alternative surrogates for MC, fetal-liver (FL) and fetal-skin (FS) derived MC have been described for in vitro studies. As early as 1983, mast-cell progenitors were identified in fetal tissue, developing in the yolk sac from gestation day 9.5 on, then forming MC progenitors in fetal livers and skin [47]. Gentek et al. applied single cell sequencing methods to decipher the fate mapping of MCs and confirmed that MC progenitors are found in fetal tissue and that in fact the majority of MC do not originate from bone marrow [3]. In light of this, we established a side-by-side culture of BMMC, FSMC and FLMC in this present study and performed comparative studies on functionality, proliferation capacity and mediator content as well as reactivity to IgE-crosslinking of the FcεRI receptor and TLR ligands.

Several culture conditions under which BMMC are generated are being used in different labs, with varying FCS concentrations and with or without SCF in the medium [29,48,49]. We generated MC from adult bone marrow, fetal liver and fetal skin, and cultured these cells in medium containing SCF and IL-3 from conditioned medium. Even if the composition is not defined in detail, the use of conditioned media is generally accepted and widely used and therefore also used by us for our analyses. Our data show that the absence of SCF from MC culture conditions causes a reduction of FcεRI and an increased MFI of CD117, indicating a shift from MC to basophils. This effect is visible as early as 12 h after deprivation of SCF. This finding is supported by Ohnmacht et al., who have shown the importance of IL-3 and SCF in the culture of MC from bone marrow progenitors, as lack of SCF favoured the generation of basophils also in their setting [50]. The highest yield of MC can be generated from BMMC, and especially the high proliferation rate allows for the vast cultivation of large numbers of BMMC. However, with both FSMC and FLMC we were able to generate yields of 1 × 10^7^–5 × 10^7^ cells/embryo, which, although the proliferation rates are low due to the higher degree of maturation, are still a sufficient quantity for repetitive in vitro experiments, making them a valuable tool for research.

Classically, MC have been characterized based on granule content and proteases they produce [51]. Our studies revealed that all three tissues, bone marrow, fetal liver, and fetal skin, give rise to MC as seen in our side-by-side cultures. However, histological staining and morphological properties indicate that FSMC are closest to tissue-derived mature CTMC whereas BMMC represent MC that are least developed/matured. Aside from the common surface markers FcεRI and CD117, mature MC express the ST2 receptor constitutively [28]. In our study we were able to show the constitutive surface expression of ST2 on MC from cultures from all three tissues and that MC-specific responses to ST2 activation by IL-33 were strong in BMMC and FSMC and less pronounced in FLMC. Thus, determining ST2 expression adds to the phenotypic and functional characterization of MC generated in vitro.

IgE-dependent activation of tissue resident MC is the dominant pathway underlying immediate type allergic reactions [52]. MC can also be activated in vitro by crosslinking the FcεRI by IgE-antigen complexes, which leads to degranulation of MC. We performed functional analyses of IgE-crosslinking of the FcεRI receptor by measuring the intracellular Ca^2+^ before and following addition of DNP-HSA to IgE-sensitized MC in the absence or presence of extracellular Ca^2^. This revealed for all three MC types unequivocal immediate Ca^2+^-influx and identical sustained activation as detected by patch-clamp experiments for the K-channel SK4. However, significantly higher release of histamine and ß-hexosaminidase from FSMC was observed upon IgE-crosslinking of the FcεRI receptor. This is in line with our transcriptome data, which showed a higher log fold expression of *Hdc,* the gene for histidine decarboxylase, and *Tpsb2*, the gene encoding for MC-tryptase in steady-state FSMC compared to BMMC. Together with the higher granule content observed in FSMC compared to BMMC, this indicates a difference in MC development or maturation dependent on the tissue of cell origin. Fetal-tissue-derived MC compared to BMMC showed a phenotype and reactions more closely resembling mature in vivo MC with FSMC being the most developed.

Data generated with murine MC are often used for translational aspects, however murine MC are different from their human counterparts, as the relative expression of IL-6 in resting murine BMMC was greater in comparison to the human cord-blood derived MC [53]. Upon stimulation of MC from all three tissue types with TLR-agonists, we found that BMMC and FLMC release the highest concentrations of IL-6 upon stimulation with Pam3Cys (TLR1/2), Pam2Cys (TLR2/6) and LPS (TLR4). Interestingly and in contrast to the histamine release upon antigen-IgE activation, FSMC secreted the lowest amounts of IL-6 into the supernatant compared to positive control stimulation by PMA/ionomycin. In recent years, research on the role of MC in health and disease has focused on the mas-coupled receptor Mrgprx2. In our transcriptome analysis, we found a higher fold change of *mrgprb2* (the mouse ortholog to mrgprx2) in both fetal-derived MC, making them a valuable tool for investigating the role of MC in skin-related diseases such as atopic dermatitis, allergic contact dermatitis and urticaria [54]. We further found a decreased fold change in *s100a8* and *s100a9* in fetal-skin derived MC. S100A8/A9 is an endogenous ligand of Toll-like receptor 4 (TLR4) and is implicated in the pathogenesis of inflammatory diseases. It was found that S100a8 reduces MC degranulation and thereby the production of the cytokines IL-6 and IL-4 in response to IgE-crosslinking in vitro [55]. The reduction in the gene expression of s100a8 in FSMC compared to BMMC might explain the increased responsiveness towards crosslinking of the FcεRI receptor by IgE–antigen complexes and the increased release of histamine and hexosaminidase, as shown in Figure 4. The main characteristics of MCs studied in this paper compared to CTMC are elaborated on in Table 2. Most of the experiments represented in this study were conducted using biological triplicates, to eliminate any batch-effects that might occur. The methods applied and presented in this study are continuously being applied in our research as the basis to ensure full maturation of in vitro generated MC, thereby repeatedly confirming our findings. 

## 5. Conclusions

The maturation of MC takes place through an accurate and on-time gene expression program, regulated by the tissue-specific and non-specific transcription factors. Therefore, MC heterogeneity is a result of the microenvironment, which educates the cells and consequently, the effector functions of MCs are different depending on the tissue in which the cell mature and differentiate [9,56,57]. Our studies showed that there are differences between MC generated in vitro from different tissues that need to be addressed, especially when MC immune functions are studied. Our study indicates that there is already an imprinting of the precursor cells for MC in fetal tissues that remains effective even during in vitro cultivation. Thus, the tissue of origin imprints MC precursor cells so that these cells, despite identical culture conditions and consecutive equal surface expression of CD117 and FcεRI, acquire distinct phenotypes and signatures. Under basal conditions, even within the same tissue, MC populations are phenotypically different and form further specific subpopulations [58]. With this study we shed light on the heterogeneity of MC used for in vitro analysis and show that fetal-derived MC may be a more appropriate surrogate for some in vitro studies, as these cells show a higher degree of differentiation and maturity and thus resemble connective tissue MC more closely. Of course, research with fetal tissue needs to adhere to local regulations, however organ retrieval is generally accepted.

## Figures and Tables

**Figure 1 cells-11-00928-f001:**
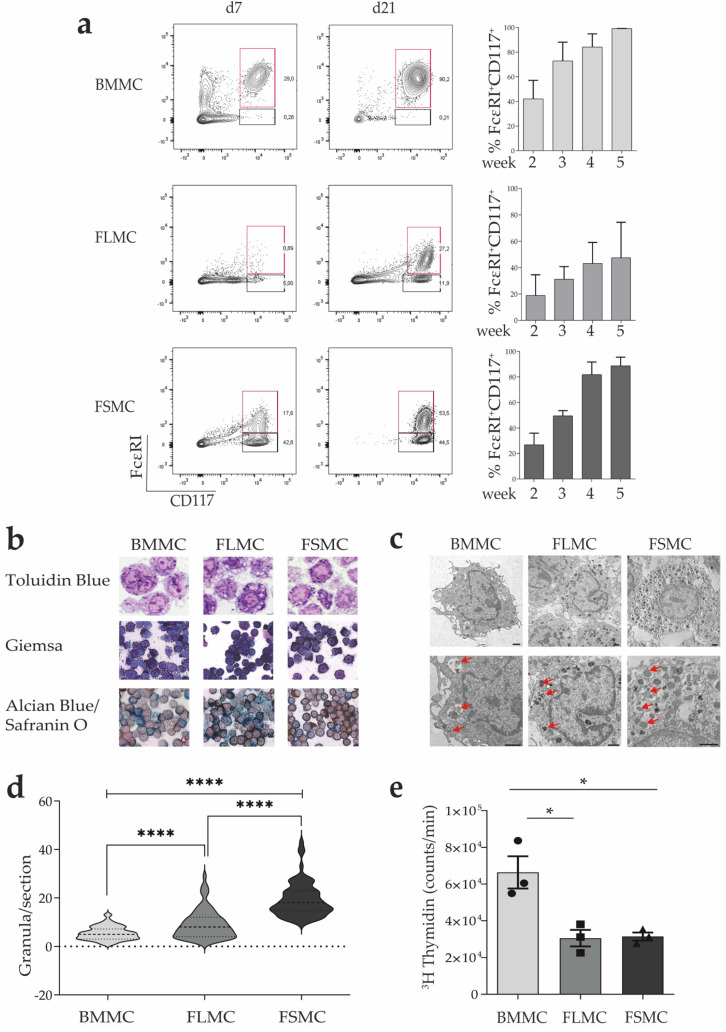
Generation and characterization of MC. (**a**) MC determined by surface markers using flow cytometry. Cells were cultured in medium with SCF and IL−3 and flow cytometry performed weakly. Mature MC were classified as LD^−^CD45.2^+^FcεRI^+^CD117^+^ cells. Bar graphs represent quantification of maturation from week 2 until week 5 (*n* = 3, mean with SEM). (**b**) Morphology of BMMC, FLMC and FSMC analyzed by histology. Semithin sections were stained with Toluidin Blue (**top panel**, 100× magnification), Giemsa (**middle panel**, 40× magnification), and Alcian Blue/Safranin O (**bottom panel**, 40× magnification). (**c**) Structural analysis of MC was conducted by electron microscopy. Representative images with red arrows representing areas with granules (scalebar 1 µM), (**d**) quantification of granules per section (*n* = 70). (**e**) Proliferation of MC based on the incorporation of ^3^H−thymidine. Proliferation rate determined as counts/min (*n* = 3, mean with SEM). Unpaired *t*-test * *p* > 0.05, **** *p* > 0.0001.

**Figure 2 cells-11-00928-f002:**
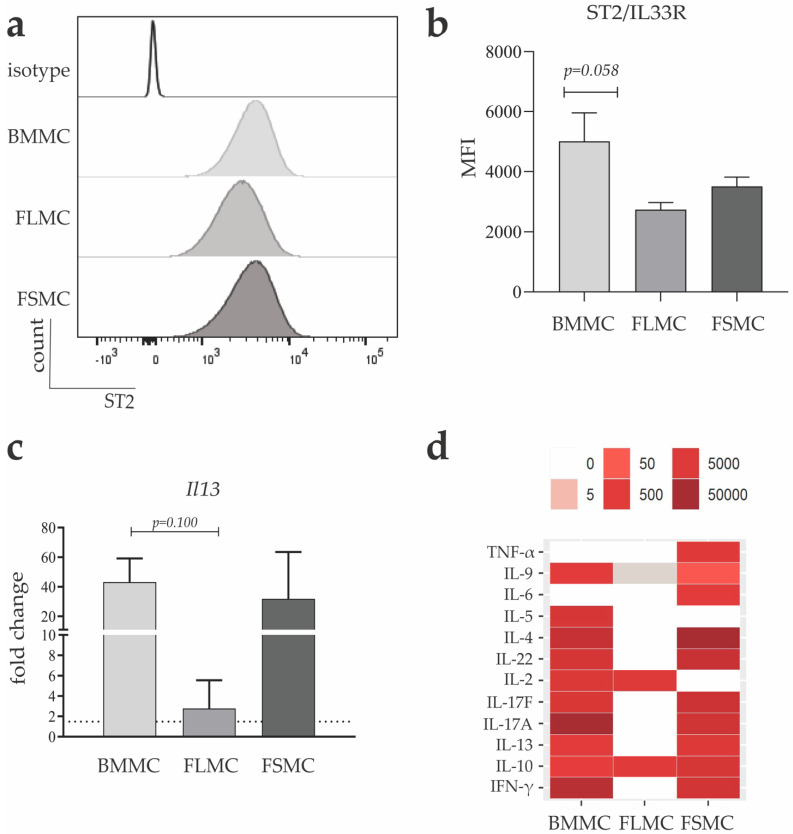
MC response to IL−33 stimulation. Flow cytometry analysis of surface receptor ST2 shown as (**a**) a representative histogram of surface expression on BMMC (light grey), FLMC (medium grey) and FSMC (dark grey) and (**b**) quantified as mean fluorescent intensity (MFI) (*n* = 4, mean with SEM). (**c**) MC were stimulated with 10 ng/mL IL−33 for 1 h and RNA was isolated. mRNA expression level of IL−13 was analyzed by quantitative real-time PCR. Fold change determined as 2^−∆∆ct^ (*n* = 3, mean with SEM). (**d**) MC were stimulated with 10 ng/mL IL−33 for 24 h and concentration of secreted type 2 cytokines were measured in the supernatant via enzyme-linked immunosorbent assay (*n* = 3, mean with SEM). Color gradient is depicted from 0–50,000 pg/mL.

**Figure 3 cells-11-00928-f003:**
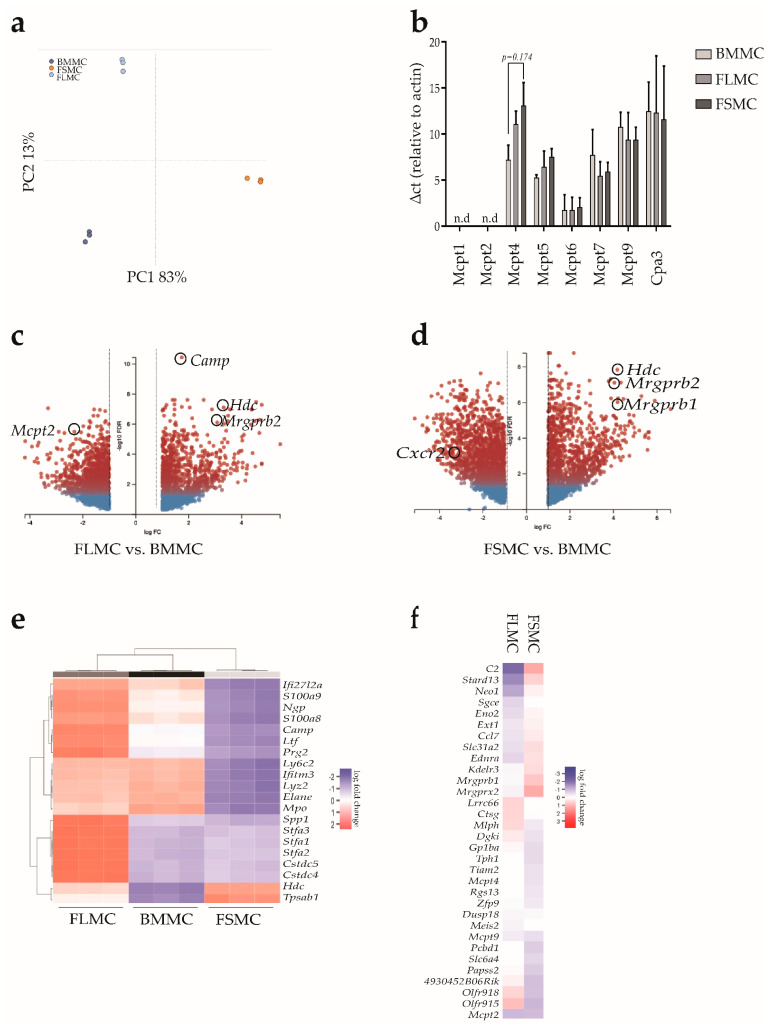
MC transcriptome analysis. (**a**) RNA sequencing of BMMC (dark blue), FLMC (light blue) and FSMC (orange) was performed. Principal component analysis showed variance among the MC types (*n* = 3). (**b**) MC protease mRNA expression analyzed by quantitative real−time PCR. ∆ct values relative to housekeeping gene β−actin (*n* = 3, mean with SEM). (**c**,**d**) Volcano plot of differentially expressed genes between the groups (**c**) FLMC vs. BMMC and (**d**) FSMC vs. BMMC analyzed with the webtool Degust (*n* = 3, FDR ≤ 0.01, abslogFc > 1). (**e**) Heatmap showing the top 20 DEG between BMMC (black), FLMC (dark grey) and FSMC (light grey). Upregulated genes are shown in red and downregulated genes in blue (*n* = 3). (**f**) Differential expression of genes associated with the MC signature by Dywer et al. [18]. Gene expression of FLMC and FSMC normalized against BMMC. Upregulated genes are shown in red and downregulated genes in blue. (*n* = 3, FDR ≤ 0.01, abslogFc > 2).

**Figure 4 cells-11-00928-f004:**
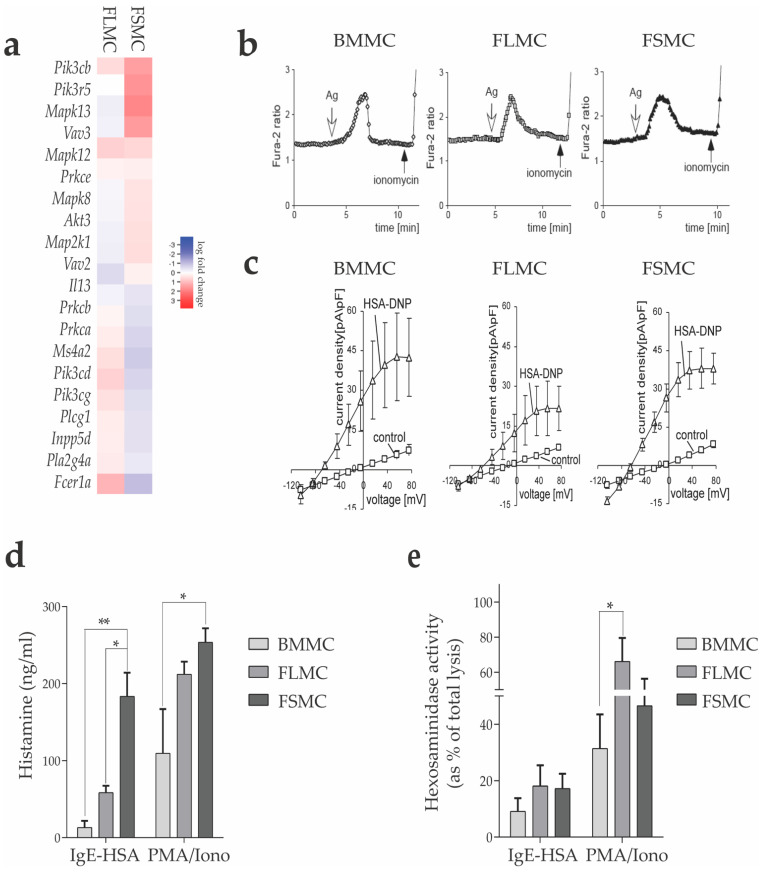
MC response to IgE stimulation. (**a**) Heatmap of differentially expressed genes relative to BMMC (FDR > 1, abs logFC > 1.5×). Gene list based on the FC_Epsilon_RI_Signaling gene set from KEGG. (**b**) MC were incubated over night with IgE anti−DNP. Prior to stimulation, cells were loaded with fluorescent Fura−2. Calcium influx was triggered by DNP−HSA. (**c**) Comparable sustained activation of the K−channel SK4 as shown by patch clamp analysis. (**d**) Histamine levels detected in supernatant after PMA/ionomycin or IgE−HSA stimulation for 24 h using an enzyme-linked immunosorbent assay (*n* = 4, mean with SEM), * *p* > 0.05 ** *p* > 0.01). (**e**) MC were stimulated with PMA/ionomycin or IgE−DNP for 30 min. Hexosaminidase activity was calculated as percent released relative to total cell lysis. (*n* = 4, mean with SEM), * *p* > 0.05.

**Figure 5 cells-11-00928-f005:**
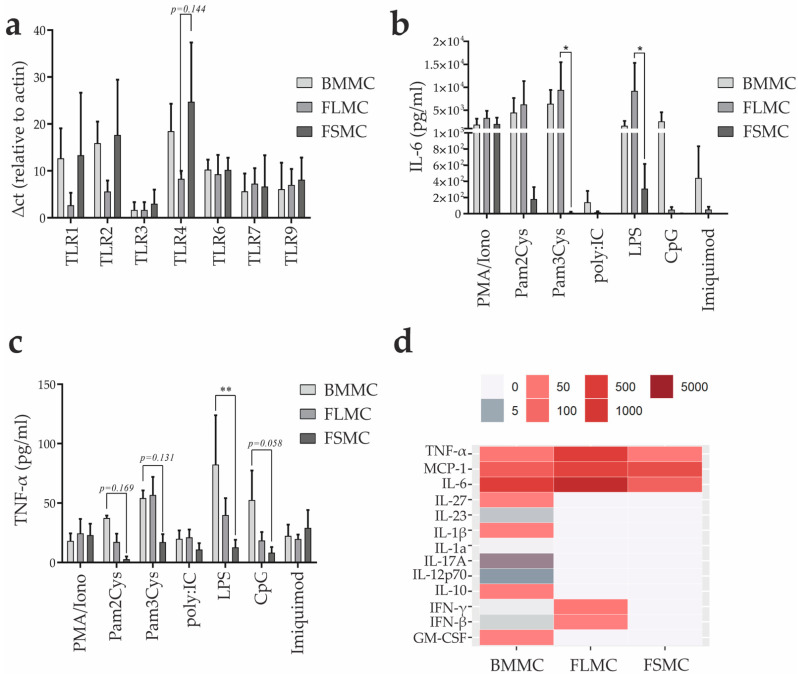
MC response towards TLR stimulation. (**a**) Toll−like receptor mRNA expression levels of MC analyzed by quantitative real-time PCR. ∆ct relative to housekeeping gene HPRT (*n* = 3, mean with SEM). (**b**,**c**) MC were stimulated with various TLR−agonists for 24 h. Concentration of secreted (**b**) IL−6 and (**c**) TNF-a were measured in the supernatant via enzyme-linked immunosorbent assay (*n* = 3, mean with SEM). (**d**) MC were stimulated with LPS for 24 h. Cytokines in the supernatant were analyzed using an enzyme-linked immunosorbent assay. Concentrations are given in pg/mL. Color gradient is depicted from 0−5000 pg/mL. * *p* > 0.05, ** *p* > 0.01.

**Table 1 cells-11-00928-t001:** Yield and maturation time for MC types.

Cell Type	Culture Time (in Weeks)	Yield
BMMC	4–6	1 × 10^8^/tibia
FLMC	6–10	5 × 10^7^/embryo
FMSC	4–6	1 × 10^7^/embryo

**Table 2 cells-11-00928-t002:** Characteristic profile of MC surrogates.

Title 1	BMMC	FLMC	FSMC	CTMC
Alcian blue/safranin staining	Blue	Blue/Red	Red	red
Toluidine staining	violet	Violet	Violet	violet
Granules	+	++	+++	+++
Histamine content	+	++	+++	+++
Proteases	Mcpt4,5,6,7,9 Cpa3	Mcpt4,5,6,7,9 Cpa3	Mcpt4,5,6,7,9 Cpa3	Mcpt4,5,6,7,9 Cpa3
Degranulation by IgE	+++	+++	+++	+++
TLR expression	TLR1,2,3,4,6,7,8	TLR1,2,3,4,6,7,8	TLR1,2,3,4,6,7,8	TLR1,2,3,4,6,7,8,9
Phenotype				
CD45	+	+	+	+
CD117	+	+	+	+
FcεRI	+	+	+	+
ST2 expression	++	+	++	++
Cytokine secretion upon LPS stimulation	+++	++	+	+++
Cytokine secretion upon IL-33 stimulation	+++	+	+++	+++

## Data Availability

The data sets generated during the current study are available from the corresponding authors on reasonable request.

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
