# Peer review of "Fetal Tissue-Derived Mast Cells (MC) as Experimental Surrogate for In Vivo Connective Tissue MC"

_cells, 2022, doi:10.3390/cells11060928_

Round 1

Reviewer 1 Report

Iuliano and co-workers have extensively and carefully investigated the morphological, ultrastructural, immunological, and molecular characteristics of mouse BMMC, FSMC, and FLMC. The results of this study demonstrate that there are several similarities and significant differences between mast cells generated in vitro from different mouse tissues. This study will be of interest to researchers in the field of mast cell biology.

Major points:

  1. In several circumstances, the Results section includes aspects that belong to the Discussion. For example, lines 295-299; lines 322-323; lines 377-383.
  2. Results, 3rd paragraph, lines 288-293. The ultrastructural characteristics of BMMC, FSMC, and FLMC are interesting. Mast cell granules are ultractructurally very heterogeneous. I wonder whether the authors had the opportunity to investigate the ultrastructure of cytoplasmic granules of different types of mast cells.
  3. Figures 1D and 1E. Apparently the results of 2/3 experiments are presented. The results presented in Figure 1D refer only to two experiments, presumably too few to draw conclusions.
  4. Section 3.2. The results presented in Figure 2D are probably not discussed in the text.
  5. Section 3.3, line 346. The authors mention that the experiments were performed with > 90% CD117+ FceRI+ cells. This raises the issue of purity of mast cells in their preparations and the possible role of contaminating cells. This aspect should be mentioned in the Discussion section.
  6. Figure 2C. Although the results probably derived from a limited number of experiments (3), statistical analysis should be performed.
  7. Figures 4D and 4E. The different presentation of the data in Fig. 4D (ng/ml) and 4E (% of total lysis) could lead of misinterpretation of the results (Lines 424-440). If possible, I would suggest to present the results in Fig. 4D as % histamine release.
  8. Figure 5 (A-B-C). Are the differences between different types of mast cells significant?
  9. Most of the data presented derived from a limited (3) number of experiments. This is a limitation that should be mentioned in the Discussion.

Minor points:

Abstract

The first sentence is too long. It can be rephrased as follow: “Bone marrow-derived mast cells are matured from bone marrow cells in medium containing 20% fetal calf serum (FCS), interleukin (IL)-3 and stem-cell factor (SCF) and are used as in vitro model to study mast cells (MC) and their role in health and disease.

Line 29. “Of note” can be omitted.

Line 29. Substitute “strong” with “highly expressed”.

Line 31. Compared to what?

Introduction

Line 39. “Importantly” can be omitted.

Lines 39-40. “mostly of surface organs arise” can be omitted.

Line 42. Perhaps, reference n.3 can be added to reference n.2.

Line 54. “Strictly” can be omitted.

Line 88. Start a new paragraph.

Line 90. “with each other” can be omitted.

Materials and methods

Line 106. “RPMI1640” should be “RPMI 1640”

Line 125. It should be “anti-CD16/CD32”.

Line 136. “toluidin” should be consistently “toluidine”.

Line 149. Please, add “at 37°C”.

Line 153. It should be “anti-DNP”.

Line 154. “over night” should be 18, 20, or 24 hours?

Line 176. Please, indicate the value (e.g., 22°C) of “room temperature”.

Results

Line 278. “Toluidine”.

Line 282. Substitute “a second” with “another”.

Lines 296-299. Perhaps, these statements belong to the Discussion section.

Line 313. This statement should be corrected. Also human basophils, neutrophils, and other immune cells express ST2.

Line 377-383. This paragraph can be shortened.

Line 399. “analyzes” should be “analyzed”.

Lines 428-434. These statements probably should be revised. See major point n. 7.

Lines 464-466. Are these statements supported by the statistical analysis of data presented in Figure 5?

Lines 543-547. These sentences are not relevant to the content of this paper.

Line 556. Please, add specific references.

Author Response

Reviewer 1

Dear Reviewer, thank you for your constructive criticism of our manuscript. Please find our comments to your remarks below. We hope that we were able to answer your pending questions to your satisfaction. Furthermore, please find that we have adjusted your minor corrections in the script.

  1. In several circumstances, the Results section includes aspects that belong to the Discussion. For example, lines 295-299; lines 322-323; lines 377-383.

Thank you for pointing this out. Although we believe that a short sentence summarizing the main findings of the figure is important to understand the incentive for conducting the following experiments, we do agree that at some points this was too lengthy. We have adjusted this (line 360-361; 401-402; 476; 544; 583-586).

  1. Results, 3rd paragraph, lines 288-293. The ultrastructural characteristics of BMMC, FSMC, and FLMC are interesting. Mast cell granules are ultrastructurally very heterogeneous. I wonder whether the authors had the opportunity to investigate the ultrastructure of cytoplasmic granules of different types of mast cells.

Thank you for this interesting hint, however, this was beyond the scope of this method paper. Therefore, we cannot comment on the nature of the granules of the MC, apart from the quantity and content, measured upon release. We do however find this aspect to be highly interesting and in-depth analysis of mast cell granules and extracellular vesicles is currently ongoing.  

  1. Figures 1D and 1E. Apparently, the results of 2/3 experiments are presented. The results presented in Figure 1D refer only to two experiments, presumably too few to draw conclusions.

Thank you for pointing this out. We have revised our data and adjusted the figures. The quantification of granules (Figure 1d) is now n=70 showing strong significant differences between the MC types. The proliferation (Figure 1E) is n=3, where each n is a biological replicate, which was further analyzed as technical triplicates in the assay.

  1. Section 3.2. The results presented in Figure 2D are probably not discussed in the text.

Thank you very much for pointing this out, this was an error on our side. We have corrected this and inserted the correct reference to Figure 2D in the text (page 9, line 402).

  1. Section 3.3, line 346. The authors mention that the experiments were performed with > 90% CD117+ FceRI+ cells. This raises the issue of purity of mast cells in their preparations and the possible role of contaminating cells. This aspect should be mentioned in the Discussion section.

Thank you for this comment. Our cells were cultured under comparable conditions. To ensure maturity, a minimum of 90% of cells were required to be CD117+ FceRI+ by flow cytometry. This means that following in vitro experiments were only conducted when this minimum was reached, however in most cases cell cultures showed a greater purity than this. Our histological experiments support that the remaining 3-10% of cells are immature mast cells, rather than impurities or contaminations.

  1. Figure 2C. Although the results are probably derived from a limited number of experiments (3), statistical analysis should be performed.

Thank you for pointing this out. We have adjusted this and inserted the statistics for this figure. Unfortunately, due to the high standard deviation, the results are not significant but show a clear trend, therefore we have inserted the p-value in the figure, to undermine our statement.

  1. Figures 4D and 4E. The different presentations of the data in Fig. 4D (ng/ml) and 4E (% of total lysis) could lead of misinterpretation of the results (Lines 424-440). If possible, I would suggest to present the results in Fig. 4D as % histamine release.

The different presentation is due to the different generation of the data. The stimulation of the MC was conducted in parallel but with different media. Figure 4E was created using a hexosaminidase-assay which uses the complete lysis of cells (achieved by Triton-X100 treatment) to determine a percentual release of hexosaminidase, as this assay does not entail a standard curve. Figure 4D is based on a commercial histamine ELISA, which used a standard curve to calculate concentrations. The total lysis of cells was not measured in this ELISA, therefore a conclusion on % histamine release cannot be made, as no reference for 100% was included.

  1. Figure 5 (A-B-C). Are the differences between different types of mast cells significant?

Thank you for pointing this out. We have revised the statistical calculations for figure 5A-C. Where applicable (Figure 5B and Figure 5C), we have incorporated the statistics (indicating with *). Furthermore, we have incorporated the p-values where applicable to strengthen the evident trend we are witnessing.

9.     Most of the data presented derived from a limited (3) number of experiments. This is a limitation that should be mentioned in the Discussion.

Our data were derived from biological triplicates (i.e. three separate weans/mice were used for the generation of the cultures). For qPCR, ELISA, etc. these biological replicates were also analyzed as technical triplicates. We took measures to eliminate the potential risk of fluctuation based on batch effects. We have made the readers of the paper aware of this fact, by incorporating this into the discussion (page 18, line 700-705).  

Reviewer 2 Report

In this article, Iuliano et al evaluated distinct parameters of maturity, mediator release and transcriptome of side-by-side cultures of murine bone marrow-derived mast cells (BMMCs), fetal liver mast cells (FLMCs) and fetal skin mast cells (FSMCs) to investigate which one could mimic the characteristics of mature connective tissue-mast cells (CTMCs).  After the analysis of histological characteristics, cell morphology, capacity of response to FceRI and Toll like receptors, calcium mobilization and massive gene transcription, authors conclude that mast cells derived from fetal tissue are the closest model to mature CTMCs.

This article defines a well-conducted study and gives light on the important question about the best in vitro model to analyze mast cell (MC) function. Since authors compare their results with those obtained by distinct research groups over the years in their effort to characterize distinct populations of MCs, their results will be very useful for distinct laboratories, specially those interested on the characteristics and function of connective tissue-resident MCs.

Major points:

1.Isolation of FSMCs and FLMCs was performed from fetuses of 16 days. Authors should explain why that time point was chosen and discuss the relevance of their findings in the light of the observations of Gentek, et al., 2018 (Immunity, 2018, 48:1160).

2.Since the distinct populations of mast cells used in this study were generated in the presence of conditioned media from other cell types (X63Ag8-653mIL-3 as a source of IL-3 and CHO cells as a source of SCF), and not with recombinant purified cytokines, participation of non-characterized ligands present on those media on the phenotypic characteristics of analyzed cells cannot be ruled out. This point must be mentioned in the discussion.

3.Authors should cite original manuscripts instead reviews in some aspects, such as the study of particular TLR receptors in MCs or the influence of microenvironment on MC phenotype.

4. Authors should consider to elaborate a table with the main characteristics of the MCs studied and compare them with the described attributes of mature connective tissue mast cells.

Minor points:

Line 101 states that C57BL/6 mice were used but no specific sub-strain (6/J or 6/N) is mentioned.  Later, on line 114, strain C57Bl/6N is mentioned. Since important differences have been reported between sub-strains of C57BL/6 mice have been reported (see Kang, SK., et al. Epilepsia Open, 2019, 4:164-169), authors should explain if different sub-strains were used for the generation of distinct populations of MCs.

Line 153 states that DNP IgE from Sigma-Aldrich was used for sensitization. Was clone SPE7 used? If this was the case, the capacity of that monoclonal antibody (specially in the high concentrations used) to induce activation of MCs and secretion of mediators during the sensitization period should be included in the discussion.

Line 132 mentions that “histological staining was performed after cells were determined matured as determined by flow cytometry”, however, in section 2.3, criteria to consider mature the cell cultures are not mentioned. Although this is explained on line 264, it should be also mentioned in the “Material and Methods” section.

Line 155 mentions that cells were stimulated with 40 ng/ml DNP-HSA but the media of stimulation is not described. Were cells resuspended on PBS or Tyrode´s/BSA buffer during stimulation or complete culture medium was utilized?  Was the same media utilized for histamine determination?

Line 156 mentions that phorbol-12-myristate-13-acetate (PMA) was used as a control, while in the same compound is mentioned as phorbol myristate acetate on line 170. Please unify the nomenclature of this compound in the whole manuscript.

On section 2.7 please describe the media in which cells were stimulated during 24 h to measure cytokine release. Also, if any protease-inhibitor cocktail was used to avoid cytokine degradation should be stated.

Line 367, avoid italics in the word “expression”

 Line 372 states that Hdc catalyzes the biosynthesis of histamine to histidine. Please correct, since histidine decarboxylase catalyzes the biosynthesis of histamine from histidine.

Line 399 please substitute “analyzes” by “analyzed”

Line 458 refers to Figure 4A. Please correct, since the figure showing TLR expression on MCs is Figure 5.

Line 469 and 476 refer to figure 5. Please correct.

Author Response

Reviewer 2

Dear Reviewer, thank you for your constructive criticism of our manuscript. Please find our comments to your remarks below. We hope that we were able to answer your pending questions to your satisfaction. Furthermore, please find that we have adjusted your minor corrections in the script.

1.Isolation of FSMCs and FLMCs was performed from fetuses of 16 days. Authors should explain why that time point was chosen and discuss the relevance of their findings in the light of the observations of Gentek, et al., 2018 (Immunity, 2018, 48:1160).

Based on the publication of Yamada et al.[1], we used fetuses of day 16. This time point allows for a good dissection of fetal skin, as in earlier stages the skin is still too soft for preparation. The research of Gentek et al. showed that mast cells (MC) are initially derived from yolk sack precursors in the embryo that are progressively replaced by definitive MC later during development. In their paper in Figure 3C and 3D, they show that the highest density of MC in CD45+ cells can be found in the E16.5 stage. The paper of Gentek et al. support the precious tool that fetus-derived MC offer for research. We have incorporated this into our discussion (page 17, line 523-530).

2.Since the distinct populations of mast cells used in this study were generated in the presence of conditioned media from other cell types (X63Ag8-653mIL-3 as a source of IL-3 and CHO cells as a source of SCF), and not with recombinant purified cytokines, participation of non-characterized ligands present on those media on the phenotypic characteristics of analyzed cells cannot be ruled out. This point must be mentioned in the discussion.

Prior to use, the conditioned media were tested in bioassays in comparison to commercially available cytokines on IL-3-dependent and SCF-dependent cell lines (FDCP-1 and TF-1, respectively) We agree with the reviewer that the conditioned medium is not characterized in detail. On the other hand, it has the advantage, that, unlike commercial cytokines, an endotoxin-contamination can be ruled out. Furthermore, since we also want to provide a practical guide for in vitro studies with mast cells, we have chosen to use conditioned media, which is a generally accepted and predominantly used approach. We have now included this in the discussion (page 17, line 633-636).

3.Authors should cite original manuscripts instead reviews in some aspects, such as the study of particular TLR receptors in MCs or the influence of microenvironment on MC phenotype.

Thank you for pointing this out. Of course, it is always of higher value to have original publications that show studies instead of reviews. We have adjusted this in the text for citations regarding TLRs (page 15, line 560) and the influence of the microenvironment (page 19, line 728).

  1. Authors should consider to elaborate a table with the main characteristics of the MCs studied and compare them with the described attributes of mature connective tissue mast cells.

We thank the reviewer for recommending to summarize our findings. We have taken your advice and added a comparative table of the main characteristics elaborated on in this study to our discussion (page 18, line 706).

[1] Yamada, N., Matsushima, H., Tagaya, Y., Shimada, S., & Katz, S. I. (2003). Generation of a large number of connective tissue type mast cells by culture of murine fetal skin cells. The Journal of investigative dermatology121(6), 1425–1432. https://doi.org/10.1046/j.1523-1747.2003.12613.x

Reviewer 3 Report

In the manuscript "Fetal tissue-derived mast cells (MC) as experimental surrogate 2 for in vivo connective tissue MC" by Iuliano and colleagues, the authors compare fetal tissue mast cells to BMMC cultures as surrogate for in vivo CTMCs. The manuscript is well-written and the findings are of interest for the field. There are a few issues that need clarification.

1) in the introduction, the rationale to explore cells of fetal origin is missing. Why fetal tissue? Is there literature that disqualifies cells from more mature origin?

2) The methods lack details regarding the animals, how many fetuses were used, how were they killed (according to regulations) etc. Data regarding amount of cells retrieved from a single fetus should be in the main manuscript, from the start of the culture to the final mast cell count.

3) Why did the author not include murine peritoneal mast cells as "positive" mature mast cell controls in all assays? Now, the FSMCs are concluded to be most "mature", but there is no direct comparison with mature tissue mast cells. The only comparison is with BMMCs, which are, as known, not as mature. 

4) In figure 1b, high magnification images should be included to be able to truely distinguish the granula, which are now not clear. In addition, the data and analysis in 1d and 1e should be expanded, an n=2 is too limited for such analysis. 

5) The proliferation data in 1e are puzzling, the discrepancy between yield and proliferation of the FSMCs is not well explained (same proliferation but better yield as compared to FLMCs). This should be addressed.

6) Sample size in figure 2b should be included, as there is no error bar. Is this an n=1? If so, this should be expanded to be able to conclude anything. If it is n=3 (as in legend), error bars should be included.

7) The discussion should include a paragraph on culture limitations regarding amount of cells per fetus, ethical considerations etc.

Author Response

Reviewer 3

Dear Reviewer, thank you for your constructive criticism of our manuscript. Please find our comments to your remarks below. We hope that we were able to answer your pending questions to your satisfaction. Furthermore, please find that we have adjusted your minor corrections in the script.

  • in the introduction, the rationale to explore cells of fetal origin is missing. Why fetal tissue? Is there literature that disqualifies cells from more mature origin?

Mast cell (MC) progenitors have been reported in a variety of hematopoietic sites and peripheral tissues in the embryo and adult (summarized in Dahlin and Hallgren, 2015[1]). Furthermore, the study of Gentek et al. 2018 unraveled the fate-mapping of MC, showing that MC progenitors arise from fetal tissue.  A special feature of MC is that they enter the tissue as precursor cells via the blood and only there develop their final differentiation1. To be able to do in vitro studies, a method is needed that provides MC in sufficient quantity. Using tissue-resident MC from mature origins such as adult skin neither provides sufficient yield for in vitro analysis nor do mature MC proliferate in vitro, therefore limiting the quantity obtainable. The generation of MC from fetal tissues is an established method that had been published before (see Yamada et al.[2] and Dywer et al.[3]) Since the intention of our study is to identify an experimental surrogate for in vivo connective tissue-type MC (CTMC) we compared the most used methods here. We have adjusted the rational for using fetal tissue cells in the introduction (page 2, line 90-93).

  • The methods lack details regarding the animals, how many fetuses were used, how were they killed (according to regulations) etc. Data regarding amount of cells retrieved from a single fetus should be in the main manuscript, from the start of the culture to the final mast cell count.

For the generation of one culture of FSMC or FLMC, the litter of one pregnant mouse was used, which on average is 5 fetuses. We have adjusted the method of euthanizing in our method section (page 3, line 138 & 150). All animal handling was conducted strictly according to regulations. As suggested by the reviewer, we have now transferred the table with the information on the yield into the main text (page 6, line 339).  

  • Why did the author not include murine peritoneal mast cells as "positive" mature mast cell controls in all assays? Now, the FSMCs are concluded to be most "mature", but there is no direct comparison with mature tissue mast cells. The only comparison is with BMMCs, which are, as known, not as mature. 

The rational of this study was to find surrogates for connective tissue mast cells (CTMC). Even though peritoneal MCs (PMC) are a mature form of MCs that can be obtained from mice, they resemble mucosal MC (MMC), which differ in some properties from CTMC. This has been also shown by the study of Dywer et al.3 who found that PMC vary severely from skin CTMC.

  • In figure 1b, high magnification images should be included to be able to truely distinguish the granula, which are now not clear. In addition, the data and analysis in 1d and 1e should be expanded, an n=2 is too limited for such analysis. 

Thank you for pointing this out. We have revised our data and edited Figure 1b, by inserting toluidine images at higher magnification, where granula can be distinguished better. We were also interested in distinguishing the granula more closely, which is why we performed electron microcopy, to show the granula on a cellular level. We have further adjusted the quantification of granules (Figure 1d) which is now n=70 for our conclusion to be supported more strongly. The proliferation (Figure 1e) is n=3, where each n is a biological replicate which was further analyzed as technical triplicates in the assay.

  • The proliferation data in 1e are puzzling, the discrepancy between yield and proliferation of the FSMCs is not well explained (same proliferation but better yield as compared to FLMCs). This should be addressed.

The difference in yield is not attributed solely to the proliferation capacity of the cells, but also to the input. The number of cells isolated from one fetal-liver is lower than from the entirety of one fetus skin, therefore a higher yield of FSMC can be achieved from one fetus, even though the proliferation capacity is comparable.

  • Sample size in figure 2b should be included, as there is no error bar. Is this an n=1? If so, this should be expanded to be able to conclude anything. If it is n=3 (as in legend), error bars should be included.

As suggested, we have revised Figure 2d and adjusted the figure. The error bars were included (mean with SEM) of n=4, as well as the p-value of the statistical analysis incorporated, to strengthen the trend we are seeing (page 10, line 403).

7) The discussion should include a paragraph on culture limitations regarding the amount of cells per fetus, ethical considerations etc.

As suggested by the reviewer, we have adjusted our manuscript and incorporated a table showing the culture time each cell type requires for maturation as well as the yield that can be expected from each culture (page 6, line 339). The highest yield can be generated from BMMC, however with both, FSMC and FLMC we were able to generate yields of 1x107-5x107 cells/fetus. We have added this statement into the discussion (page 17, line 642-646). Ethical considerations should precede all experiments - whether with primary cells, cell lines or animal experiments. In our opinion, this also includes optimal cell culture conditions, which is precisely the basis of our study, namely, to make the best possible choice of cells for the particular research question. Of course, research using fetal tissue must adhere to local regulations, however, organ retrieval is always possible. We have mentioned this statement in the discussion (page 19, line 739-741).

[1] Dahlin, J. S., & Hallgren, J. (2015). Mast cell progenitors: origin, development and migration to tissues. Molecular immunology63(1), 9–17. https://doi.org/10.1016/j.molimm.2014.01.018

[2] Yamada, N., Matsushima, H., Tagaya, Y., Shimada, S., & Katz, S. I. (2003). Generation of a large number of connective tissue type mast cells by culture of murine fetal skin cells. The Journal of investigative dermatology121(6), 1425–1432. https://doi.org/10.1046/j.1523-1747.2003.12613.x

[3] Dwyer, D. F., Barrett, N. A., Austen, K. F., & Immunological Genome Project Consortium (2016). Expression profiling of constitutive mast cells reveals a unique identity within the immune system. Nature immunology17(7), 878–887. https://doi.org/10.1038/ni.3445

Reviewer 4 Report

Iuliano et al. conducted comparative studies to identify the best surrogates for mature connective-tissue mast cells (CTMCs) using BMMC, fetal skin MC (FSMC), and fetal liver MC (FLMC) models. Additionally, the authors gain some insight into the phenotypic markers, proliferation, and mediator content as well as response to mitotic activation, IgE-crosslinking of the FcεRI receptor, and innate stimulation of all types of studied MC types. These data show differences between the three MC types at the gene expression level and in regard to mediator release. Authors pointed out that the fetal-skin derived MCs are the best for analysis aiming to elucidate the role of MC in allergic inflammation and atopic diseases. Overall, the entire article is of interest. There are a few minor points (especially regarding the Methods section) that remain to be addressed, in my opinion.

  1. The concept of the Introduction provides too much unnecessary information for the understanding of the study. Therefore, I strongly suggest improving this section and substantially shortening it.
  2. Can authors explain whether they conduct their studies using males or females animals?
  3. 'Flow Cytometry for Mast Cell Surface Marker' section: Specifications of primary and secondary antibodies should be precise. There is also a lack of information regarding antibody dilution.
  4. Authors should show isotype staining for immunofluorescence stainings.
  5. A detailed composition of the PCR mixture should be provided.
  6. The sensitivity of ELISA assays should be provided.

Author Response

Reviewer 4

Dear Reviewer, thank you for your constructive criticism of our manuscript. Please find our comments to your remarks below. We hope that we were able to answer your pending questions to your satisfaction. Furthermore, please find that we have adjusted your minor corrections in the script.

  1. The concept of the Introduction provides too much unnecessary information for the understanding of the study. Therefore, I strongly suggest improving this section and substantially shortening it.

Dear Reviewer, of course our introduction is very basic and rather a recapitulation of mast cell (MC) research for any MC specialist. However, this is a technical method paper, which is directed on the one hand towards MC specialists, but also towards non-MC researchers that are starting their research in this field. Therefore, we constructed our introduction to show the origin of mast cells, and that in fact bone-marrow is not the major source of MC as has been believed for a long time and to shed light on the different receptors and functions of the cell, focusing only on those that were analyzed during this study. We do however agree that the introduction was quite lengthy, which is why we have shortened the introduction as suggested and have elaborated on the prerequisite for using fetal-derived tissue (page 2, line 90-93).

  1. Can authors explain whether they conduct their studies using males or females animals?

This is certainly an interesting question, but we did not address the potential impact of sexual hormones on MC differentiation. We have used both, male and female mice for the generation of the different mast cell types, since only in well-founded cases exclusively animals of one sex may be used.

  1. 'Flow Cytometry for Mast Cell Surface Marker' section: Specifications of primary and secondary antibodies should be precise. There is also a lack of information regarding antibody dilution.

All antibodies that were used for flow cytometry are primary antibodies coupled to specific fluorochromes (adjusted on page 3, line 161). We have made the experience that the suitable antibody dilutions can vary depending on the flow cytometer used for analysis. Therefore, we incorporated the clones used in the method section and recommend the exact clone of the antibody and conduct a titration depending on the machine used, for optimal dilution for the appropriate research lab.

  1. Authors should show isotype staining for immunofluorescence stainings.

We thank the reviewer for this important comment. We have now added the isotype-control, as recommended to Figure 2a (page 10, line 403) as well as the remaining isotypes for FCeRI, CD117 and CD49b to supplementary figure 2.

  1. A detailed composition of the PCR mixture should be provided.

For RNA isolation we have used a commercial isolation kit (PeqGold RNA Isolation Kit, VWR) and the following reverse transcription was also performed using the commercial iScript cDNA Synthesis Kit (Bio-Rad). We have followed standard protocol for the kits and have adjusted the methods sections by incorporating the concentrations of RNA used for DNAse digestion, and the concentrations of oligo-dT and dNTPs used for reverse transcription (page 5, line 271-275).

  1. The sensitivity of ELISA assays should be provided.

For all ELISA we have used commercial kits that have an incorporated standard curve. For the single cytokine ELISAs (IL-6, IL-10 and TNF-a), we have incorporated the detection limits of each kit into the methods section (page 4, line 219-223). Where appropriate supernatants were diluted prior to measuring the ELISA, to ensure that the optical density of the samples never exceeds the higher concentration of the standard curve. For the Inflammation Panel and the mouse TH Cytokine Panel (Legendplex, Biolegend), the cytokines were carefully tested and selected by the company and detection limits vary based on each specific cytokine.

Round 2

Reviewer 1 Report

The authors have answered most of my queries and they have improved the text.

The current version of the manuscript can be accepted in its current form.